# Building Social World Models with Large Language Models

**Haofei Yu** [1]  **Yining Zhao** [1]  **Guanyu Lin** [2]  **Jiaxuan You** [1]

 Code: https://github.com/ulab-uiuc/social-world-model
 Data: https://huggingface.co/datasets/ulab-ai/swm-bench

## Abstract

Understanding and predicting how social beliefs evolve in response to events—from policy changes to scientific breakthroughs—remains a fundamental challenge in social science. Given LLMs' commonsense knowledge and social intelligence, we ask: Can LLMs model the dynamics of social beliefs following social events? In this work, we introduce the concept of the Social World Model (SWM), a general framework designed to capture how social beliefs evolve in response to major events. SWM learns state-transition functions for social beliefs by mining temporal patterns in social data and optimizing the evidence lower bound, without the need for explicit human annotations linking events to belief shifts, or for expensive census data. To evaluate SWM, we introduce a benchmark, SWM-Bench, derived from real-world prediction markets, specifically Kalshi and Polymarket. SWM-Bench includes over 12k data points for social belief prediction tasks spanning diverse domains such as politics, finance, and cryptocurrency. Our experimental results show that SWM significantly outperforms time-series foundation models, achieving state-of-the-art results on Kalshi data and demonstrating competitive performance on Polymarket data, while offering interpretable insights into the underlying mechanisms of social belief dynamics.

## 1. Introduction

Diverse social beliefs shape different human communities and the future of mankind (Greif, 1994; Bar-Tal, 2000; Zou

---
[1]University of Illinois Urbana-Champaign [2]Carnegie Mellon University. Correspondence to: Haofei Yu <haofeiy2@illinois.edu>.

*Proceedings of the 43rd International Conference on Machine Learning*, Seoul, South Korea. PMLR 306, 2026. Copyright 2026 by the author(s).

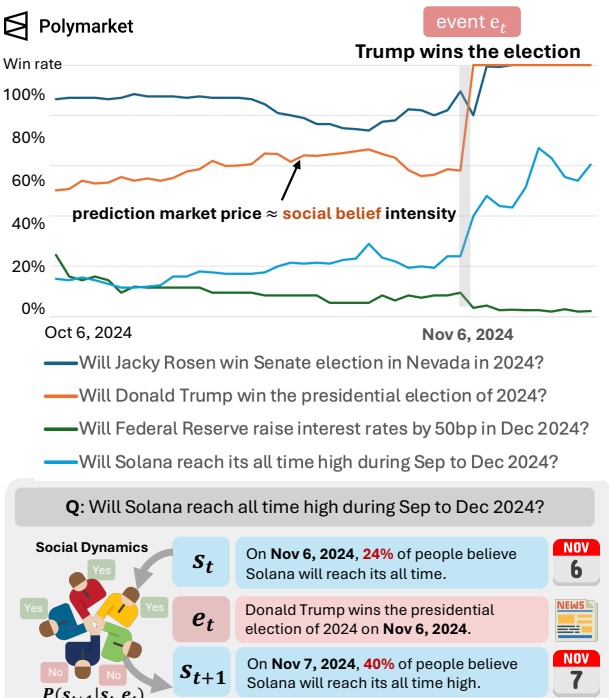

*Figure 1.* **Social events shape future social beliefs**. (**Top**) Each line tracks one social belief over time, derived from real-world Polymarket data. A breaking event triggers a sudden shift across multiple social beliefs. (**Bottom**) An example illustrating how social events directly or indirectly drive changes in social beliefs. The SWM aims to predict how these beliefs evolve based on historical states and a (hypothetical) social event.

et al., 2009). Examples of impactful social beliefs include whether Artificial General Intelligence (AGI) will emerge within the next five years (Feng et al., 2024) or who will be the next U.S. president (Barberá González & Lohan, 2024). While some widely accepted beliefs are unlikely to drastically change, other social beliefs are more volatile, shifting dramatically in response to societal events (Chang et al., 2025; Ma et al., 2025; Quinn et al., 2022). For example, as is shown in Figure 1, the U.S. presidential election results influence public expectations about the Federal Reserve's policy in December 2024, the price of Solana (a cryptocurrency), and other political events (Song et al., 2024; Tourangbam,

2024). Understanding how social beliefs evolve is crucial for a wide range of societal applications, from forecasting social events (Wang et al., 2024b; Deng et al., 2019) to improving decision-making in business and economics (Ariely, 1998; Rasheed et al., 2022; Ausat, 2023; Affeldt et al., 2017; Chen et al., 2025). This naturally leads to the question: *Can LLMs be used to model the dynamics of social beliefs in response to events?*

Modeling the dynamics of social beliefs involves three hierarchical challenges that necessitate a fundamental shift in approach: (**C1**) Quantifiability and data scarcity. The primary hurdle lies in the measurement of social beliefs. Unlike physical states, collective convictions are semantic-driven and difficult to capture as structured data (Lüders et al., 2023). This lack of high-fidelity, time-series data to represent such social beliefs makes it nearly impossible to establish standardized benchmarks for evaluating how they evolve in response to real-world stimuli (Mou et al., 2024; Song et al., 2025). (**C2**) Semantic complexity of social transitions. Even with data, the mechanisms governing social shifts are non-symbolic and semantic-driven (Simoens et al., 2022). Social dynamics are driven by nuanced psychology and cultural context rather than explicit, codified laws (Takata et al., 2025). Consequently, traditional statistical or symbolic models often fail to capture the rich, implicit transition rules that dictate how a community reacts to a novel event (Yang et al., 2025b). (**C3**) Lack of explicit attribution labels. Finally, even if we observe a belief shift, the relationship between a specific event and that shift is often obscured. Without explicit supervision (i.e., "event-to-shift" labels), models struggle to learn the mechanism behind the social belief dynamics (Bauer et al., 2023).

To address these challenges, we introduce the Social World Model (SWM), a generative framework that characterizes social belief dynamics through a state-transition paradigm, $P(\mathbf{s}_{t+1} \,|\, \mathbf{s}_t, e_t)$. Conceptually analogous to the definition of the classical world model (Bruce et al., 2024; Huang et al., 2024; Baldassarre et al., 2025),which predict future states of a physical environment from current observations and actions, SWM treats collective belief as semantic states and social events as exogenous actions. The realization of SWM involves three key technical breakthroughs: First, we ground the "state" of social beliefs in prediction market data (**C1**). We establish SWM-Bench, the first benchmark curated from high-fidelity prediction market data (covering 3k+ markets from both Polymarket[1] and Kalshi[2]). By treating market fluctuations as a proxy for collective belief, we transform latent social beliefs into a measurable and high-fidelity time-series format, providing the necessary ground truth for evaluating belief evolution. Second, we

---

[1]We refer to `https://polymarket.com/`.
[2]We refer to `https://kalshi.com/`.

utilize LLMs as the "transition engine" to navigate semantic complexity (**C2**). To capture the implicit rules of social dynamics, SWM utilizes LLMs as the backbone. By leveraging their vast pre-training on human discourse, LLMs serve as the cognitive engine, providing the commonsense reasoning and social knowledge required to simulate how complex contexts drive belief shifts (Park et al., 2023; Yan et al., 2025). Finally, we introduce a posterior-guided mechanism to bridge the attribution gap (**C3**). Since explicit labels linking specific events to belief shifts are unavailable, we utilize a posterior distribution—informed by the observed future shift $\mathbf{s}_{t+1}$—to provide high-fidelity training signals. This "posterior guidance" enables the model to perform inverse inference, effectively collecting pseudo-labels for attribution and empowering SWM with counterfactual reasoning capabilities.

**Contributions**. Empirical evaluations on SWM-Bench demonstrate that our proposed SWM achieves state-of-the-art results on the Kalshi dataset, yielding a 4% improvement in Directional Accuracy over baselines such as GPT-5.5, and demonstrates competitive performance on the Polymarket dataset. Our core contributions are three-fold: (**1**) We establish SWM-Bench, one of the first evaluation benchmarks curated from real-world prediction market data that transforms social beliefs into a quantifiable, time-series format; (**2**) We introduce the social world model framework, which leverages an LLM-driven architecture to capture the semantically driven, implicit transition rules of social dynamics; (**3**) We propose a posterior-supervised training paradigm that solves the attribution labeling problem by decoupling social reasoning from dynamics modeling. Together, these contributions provide a new pipeline for modeling and predicting the evolution of collective consensus in society.

## 2. Related Works

**Social event forecasting**. Event prediction traditionally forecasts future occurrences from historical data (Hendrycks et al., 2021; Jin et al., 2020; Deng et al., 2020; Gu, 2021) using semantic and time-series modeling (Zhang & Ning, 2024; Zou et al., 2022). Recently, LLMs have emerged as effective forecasters due to their strong social reasoning (Halawi et al., 2024; Abolghasemi et al., 2024; Schoenegger & Park, 2023), prompting new benchmarks for LLM agents (Ye et al., 2024; Karger et al., 2025). Our social world model pursues a fundamentally different objective. Rather than forecasting an event's final outcome (Woo et al., 2024; Lee et al., 2025b)—which is often unpredictable for sudden shocks like natural disasters or political shifts (Hendrycks et al., 2021)—we predict the ensuing trajectory of public opinion. Since collective reactions are more tractable to model than the events themselves, they provide a much more reliable forecasting target.

**Social simulation**. A natural approach to modeling such collective reactions is social simulation. Mainstream methods rely on LLM-driven agent-based modeling to derive emergent macro-patterns from individual micro-decisions (Gao et al., 2023; Zhang et al., 2025b; Lorig et al., 2021; Piao et al., 2025; Yang et al., 2024), often emphasizing theory-of-mind modeling (Zhou et al., 2025a) or resource distribution (Zhang et al., 2025a). A recent line of work explicitly frames this paradigm as a "social world model" (Zhou et al., 2025b; Zhang et al., 2026). Our work diverges from these agent-centric approaches in two key ways. First, methodologically: rather than attempting to recover macro-dynamics by simulating agents one by one, we parameterize the macro-level social dynamics directly. This sidesteps the high computational cost and brittle calibration associated with per-agent simulation. Second, in application: whereas prior social world models focus on micro-level behavioral or interactive tasks, we focus purely on macro-level forecasting. By representing collective belief as the state and external events as actions, our model provides a streamlined framework tailored to high-stakes, data-rich domains like prediction markets.

## 3. Preliminaries

**Data selection for social state description**. Measuring the social state is a critical challenge because traditional social-opinion sources are often noisy, vague, and unrepresentative. Survey instruments suffer from response and participation biases, while social-media corpora reflect self-selected user populations that systematically diverge from the broader public (Giorgi et al., 2022; Nedungadi et al., 2025; Javed & Kamal, 2018). Our insight is that prediction markets—where rational individuals trade financial stakes based on their expectations—serve as a higher-quality aggregated signal of public opinion (Galam, 2016). We therefore utilize data from Polymarket and Kalshi, the two leading prediction platforms[3], which provide a broad and high-resolution signal for tracking social beliefs. Three key features make these markets particularly suited for analyzing social dynamics: (**1**) Uncertainty constraint: markets inherently form around uncertain outcomes, filtering out trivial or universally agreed-upon knowledge; (**2**) Diversity and scale: a broad participant base ensures that price shifts effectively track changes in societal consensus (Arrow et al., 2008); (**3**) Investment-driven quality: financial stakes ensure deliberate decision-making, allowing prices to approximate the mean beliefs of traders and providing a microfoundation for treating them as aggregated probabilities (Wolfers & Zitzewitz, 2006). Consequently, integrating Polymarket and Kalshi

provides an exemplary foundation for modeling dynamic belief shifts (Ottaviani & Sørensen, 2007).

**Social belief**. A social belief represents a collective opinion on a binary (yes/no) proposition $q$ that resolves by a future time $T$. Formally, at time $t < T$, a social belief is denoted by the pair $b_t = (q, v_t)$, where $q$ is the textual description of the proposition and $v_t \in [0, 1]$ is the market-implied probability of a "Yes" resolution. We sample $v_t$ daily, using the closing price as the observed value. For instance, the question "Will OpenAI release GPT-5 in February 2025?" serves as $q$, and its corresponding "Yes" price serves as $v_t$.

**Social event**. A social event is a news-reported real-world occurrence that may drive belief evolution. We represent each valid non-null event as a tuple $e_t^i = (c_t^i, t)$, where $c_t^i$ is a natural-language news description of the occurrence and $t$ indexes the transition interval. To account for days without significant external shocks, we also define the null event, denoted equivalently as $e_t^0 \equiv e_t^\emptyset$.

**State space**. To capture the temporal context required for modeling social dynamics, we define the social state $\mathbf{s}_t$ for a given proposition as an *ordered* window of its most recent daily values up to day $t$: $\mathbf{s}_t = (q, (v_{t-k}, \ldots, v_{t-1}, v_t)) \in \mathcal{S}$. Here, $\mathcal{S} = \mathcal{Q} \times [0, 1]^{k+1}$, $\mathcal{Q}$ is the space of propositions, and $k$ denotes the look-back window size. Defining the state as a trajectory rather than a static point allows the model to account for momentum and volatility.

**Event space**. Because all social events can be considered as open-ended natural-language objects, the potential influences on a social belief vary dynamically. For each transition from day $t$ to day $t+1$, we collect a event candidate set $\mathcal{E}_t = \{e_t^\emptyset, e_t^1, \ldots, e_t^{m_t}\}$. This set serves as the realized event space for this transition, functioning as the externally supplied action space in our world-model formulation. While a single day may contain multiple candidate events, each forward pass of the model conditions on one specific event from this space.

## 4. Proposed Concept: Social World Model

The social world model predicts how a social belief state $\mathbf{s}_t \in \mathcal{S}$ evolves in response to a social event $e_t^i \in \mathcal{E}_t$. Unlike standard time-series forecasting, which typically relies on historical trends alone, our framework explicitly models discrete social events as drivers of state transitions. This approach captures the temporal relationship between news-reported real-world occurrences and shifts in public opinion, enabling the simulation of belief dynamics under both observed and hypothetical scenarios. For instance, given the historical trajectory of an election forecast, the social world model can predict how that belief state would shift following a hypothetical policy change.

**Definition 4.1** (*Social World Model*)**.** Let $\mathcal{S}$ be the state

---

[3]Kalshi and Polymarket form a *de facto* duopoly, generating more than $44B in trading volume in 2025 and jointly accounting for 85–90% of total prediction-market volume (Park, 2025); the precise share fluctuates week to week as the ecosystem evolves.

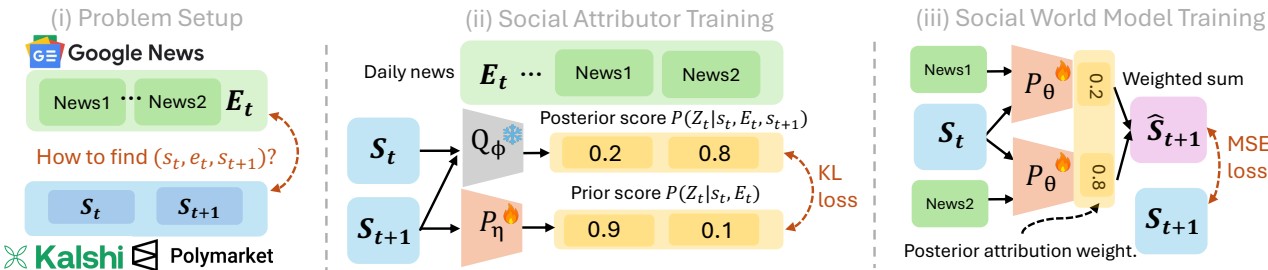

*Figure 2.* **Overview of the social world model training framework.** Our architecture employs LLMs as the backbone for three core modules: the social attributor ($P_\eta$), the posterior-guided social attributor ($Q_\phi$), and the social world model ($P_\theta$). Notably, only $P_\eta$ and $P_\theta$ are updated during training. The training pipeline proceeds in three steps: (**i**) collecting observational state-event transitions $(\mathbf{s}_t, e_t^i, \mathbf{s}_{t+1})$; (**ii**) optimizing the social attributor $P_\eta$ via KL divergence against the posterior $Q_\phi$; and (**iii**) performing posterior-guided training on the SWM $P_\theta$ using the attributed events.

space of social belief trajectories and let $\mathcal{E}_t$ be the event space at time $t$. A social world model is defined as an event-conditioned transition distribution:

$$\mathbf{s}_{t+1} \sim P_\theta\big(\cdot \mid \mathbf{s}_t, e_t^i\big), \qquad \mathbf{s}_t \in \mathcal{S}, \ e_t^i \in \mathcal{E}_t, \quad (1)$$

where $\theta$ denotes the model parameters.

In practice, since the proposition $q$ is fixed within a trajectory, the transition model can be implemented by predicting the next belief value $v_{t+1}$ and deterministically updating the state window as $\mathbf{s}_{t+1} = (q, (v_{t+1-k}, \ldots, v_{t+1}))$. Under the null event $e_t^\emptyset$, the model captures endogenous belief evolution in the absence of external shocks; conditioning on a non-null event yields an event-conditioned "what-if" estimate of that event's marginal effect.

**Comparison to existing world-model definitions**. Classical world models (Hu & Shu, 2023) typically assume a Markovian transition $\mathbf{s}_{t+1} \sim P_\theta(\mathbf{s}_{t+1} \mid \mathbf{s}_t, a_t)$, where $\mathbf{s}_t$ is a physical observation (e.g., a video frame) and $a_t$ is an agent-controlled action. Our formulation differs in three key aspects: (**1**) *States as belief trajectories*: A single instantaneous value is insufficient because social beliefs depend on recent history, momentum, and volatility. By encoding a historical window $(v_{t-k}, \ldots, v_t)$ into the state, we approximate a Markovian state representation at the trajectory level while supplying the temporal context necessary to interpret an event. (**2**) *Events as exogenous shocks*: Unlike the agent-controlled actions $(a_t)$ found in robotics or gaming, $e_t^i$ acts as an exogenous shock that perturbs the belief state. A designated null event captures periods with no observed external driver. (**3**) *Shared dynamics*: Rather than modeling a single market, we treat each proposition or market as an instance in $\mathcal{S}$ and learn a universal transition function $P_\theta$. This leverages the LLM's broad social reasoning and generalization capabilities to predict how community beliefs react to events. By sharing parameters across domains, the model transforms sparse, per-market data into a joint learning problem, replacing thousands of isolated forecasts with a single world model.

## 5. Building a Social World Model with LLMs

Following Sec. 4, we train SWM on observational trajectories where each training example is a belief transition $(\mathbf{s}_t, \mathcal{E}_t, \mathbf{s}_{t+1})$. To move beyond traditional time-series forecasting, our goal is to capture the underlying *mechanisms* of social dynamics, specifically how external events drive collective belief shifts. However, the true social event responsible for a social belief shift is latent and often entangled in complex information streams. We therefore cast learning as a latent event attribution task, supervised by posterior LLM hindsight. To render these mechanistic dynamics tractable from observation, we rely on two simplifying assumptions:

**(A1) First-order event approximation**. Although a belief shift may result from multiple interacting events, we approximate each transition by conditioning on a single hypothesized event $e$: $P_\theta(\mathbf{s}_{t+1} \mid \mathbf{s}_t, e)$. The learned attribution is thus a soft distribution over which single event best explains the transition, rather than a full decomposition over event combinations. This approximation naturally fits prediction markets, where large price movements typically align with a single salient public event.

**(A2) Conditional event exogeneity**. To interpret event-conditioned predictions causally, we assume candidate events are conditionally exogenous given the current belief state $\mathbf{s}_t$. This assumption is robust for scheduled announcements and external news, though weaker for reflexive, market-generated events. When exogeneity fails, SWM should be viewed as a predictive model rather than a causally identified one. In most prediction market scenarios, however, this assumption holds.

### 5.1. Modeling with Latent Event Attribution

We model the driving event for each transition as a categorical latent variable $Z_t \in \{0, 1, \ldots, m\}$. This variable indexes the $m$ candidate events in the event set $\mathcal{E}_t$, with $Z_t = 0$ reserved for the null event $e_t^\emptyset$. Conditioned on $Z_t = i$, the transition is governed by the social world model $P_\theta(\mathbf{s}_{t+1} \mid \mathbf{s}_t, e_t^i)$, which depends exclusively on the se-

lected event rather than the full set $\mathcal{E}_t$. A prior attributor $P_\eta(Z_t = i \mid \mathbf{s}_t, \mathcal{E}_t)$ scores each candidate based on the context available prior to the shift. Marginalizing over $Z_t$ yields the predictive distribution:

$$P(\mathbf{s}_{t+1} \mid \mathbf{s}_t, \mathcal{E}_t) = \sum_{i=0}^{m} \underbrace{P_\theta(\mathbf{s}_{t+1} \mid \mathbf{s}_t, e_t^i)}_{\text{social world model}} \underbrace{P_\eta(Z_t{=}i \mid \mathbf{s}_t, \mathcal{E}_t)}_{\text{prior attributor}}.$$
(2)

During inference, $P_\eta$ supplies the event weights; during training, it is supervised by a hindsight posterior (Sec. 5.2).

**The null event**. The null event completes the attributor's label space. When no candidate adequately explains the transition, the hindsight posterior concentrates its mass on $Z_t = 0$ to form the latent variable. Consequently, intervals lacking a clear driving event still provide supervision for $P_\eta$ rather than being discarded. The null branch requires no learnable parameters: we fix $\mathbb{E}_{P_\theta}[\mathbf{s}_{t+1} \mid \mathbf{s}_t, e_t^\emptyset] = \mathbf{s}_t$, representing the martingale forecast of an efficient market. Because the null branch is parameter-free, transitions attributed to the null event supervise only $P_\eta$ and contribute no gradient to $\theta$; thus, $\theta$ exclusively models the dynamics of non-null events. This persistence baseline also serves as the reference for measuring event effects: under assumption (A2), $\mathbb{E}_{P_\theta}[\mathbf{s}_{t+1} \mid \mathbf{s}_t, e] - \mathbf{s}_t$ represents the abnormal effect of $e$, akin to classical event studies (Fama et al., 1969); without (A2), it simply denotes a predictive deviation.

## 5.2. Training with Posterior Guidance

Directly maximizing the marginal log-likelihood of Eq. (1) is intractable: $Z_t$ is unobserved, and most candidates in $\mathcal{E}_t$ are irrelevant to any given shift. Our key observation is that attribution is far easier in *hindsight*. A frozen LLM that sees the realized outcome—the posterior attributor $Q_\phi(Z_t \mid \mathbf{s}_t, \mathbf{s}_{t+1}, \mathcal{E}_t)$—can reliably judge which candidate's timing and content explain the observed shift, producing a sharply peaked distribution $\pi_t := Q_\phi(\cdot \mid \mathbf{s}_t, \mathbf{s}_{t+1}, \mathcal{E}_t)$ (see Appendix C.2). The forward models, which must act before the outcome is revealed, are trained to match these hindsight labels: $P_\theta$ learns the transition dynamics conditioned on the attributed event, and $P_\eta$ learns to predict the attribution without access to $\mathbf{s}_{t+1}$. The procedure is thus a form of pseudo-labeling: hindsight supplies the labels, and foresight learns from them.

**Training objective**. We minimize:

$$\mathcal{L}_{\theta,\eta} = \underbrace{-\,\mathbb{E}_{Z_t \sim \pi_t}\big[\log P_\theta(\mathbf{s}_{t+1} \mid \mathbf{s}_t, e_t^{Z_t})\big]}_{\mathcal{L}_{\text{wm}}} + \underbrace{D_{\text{KL}}\big(\pi_t \| P_\eta\big)}_{\mathcal{L}_{\text{attr}}}.$$
(3)

The two terms have a simple reading. $\mathcal{L}_{\text{wm}}$ trains the world model on each transition paired with its hindsight-attributed event, and $\mathcal{L}_{\text{attr}}$ distills the hindsight attribution into the forward attributor. Since $P_\theta$ and $P_\eta$ are separate models with disjoint parameters, the two terms decouple and are

optimized independently, requiring no balancing hyperparameter. Because $\pi_t$ is highly concentrated in practice (Figure 7), we evaluate $\mathcal{L}_{\text{wm}}$ only over its top-$k$ support with renormalized weights $\bar{\pi}_t$. The full procedure is outlined in Algorithm 1.

**Algorithm 1** Social World Model Training Algorithm

**Require:** Social belief trajectories $\{\mathbf{s}_{0:T}\}$; event candidate sets $\mathcal{E}_t$; learning rates $\alpha_\theta, \alpha_\eta$; sparsity cutoff $k$
**Require:** Social world model $P_\theta$, prior attributor $P_\eta$; frozen posterior attributor $Q_\phi$
**Ensure:** Trained parameters $(\theta, \eta)$
 1: **for** each trajectory $\mathbf{s}_{0:T}$ **do**
 2:     **for** $t = 0, \ldots, T-1$ **do**
 3:         $\pi_t \leftarrow Q_\phi(\cdot \mid \mathbf{s}_t, \mathbf{s}_{t+1}, \mathcal{E}_t)$
 4:         $\bar{\pi}_t \leftarrow \text{top-}k\text{-support}(\pi_t)$, with support $\mathcal{I}_t$
 5:         $\mathcal{L}_{\text{wm}} \leftarrow -\sum_{i \in \mathcal{I}_t} \bar{\pi}_t^i \log P_\theta(\mathbf{s}_{t+1} \mid \mathbf{s}_t, e_t^i)$
 6:         $\mathcal{L}_{\text{attr}} \leftarrow D_{\text{KL}}(\pi_t \| P_\eta(\cdot \mid \mathbf{s}_t, \mathcal{E}_t))$
 7:         $\theta \leftarrow \theta - \alpha_\theta \nabla_\theta \mathcal{L}_{\text{wm}}$
 8:         $\eta \leftarrow \eta - \alpha_\eta \nabla_\eta \mathcal{L}_{\text{attr}}$
 9:     **end for**
10: **end for**
11: **return** $(\theta, \eta)$

**World model term**. Since the belief change $\Delta_t = \mathbf{s}_{t+1} - \mathbf{s}_t$ is continuous, we adopt the simplest continuous likelihood: a homoscedastic Gaussian $\Delta_t \sim \mathcal{N}(\mu_\theta(\mathbf{s}_t, e), \sigma^2 I)$, where $\mu_\theta$ is a regression head on the LLM backbone. The variance is fixed because downstream use (Sec. 5.3) consumes only point predictions; this reduces the negative log-likelihood, up to constants, to a weighted squared error:

$$\mathcal{L}_{\text{wm}} = \sum_{i \in \mathcal{I}_t} \bar{\pi}_t^i \left\| \Delta_t - \mu_\theta(\mathbf{s}_t, e_t^i) \right\|^2,$$
(4)

i.e., each credible candidate's predicted change is regressed toward the realized $\Delta_t$, weighted by its hindsight probability. For the null branch we fix $\mu_\theta(\mathbf{s}_t, e_t^\emptyset) \equiv 0$, so $\theta$ models only non-null events, and future states are recovered via $\widehat{\mathbf{s}}_{t+1} = \mathbf{s}_t + \mu_\theta(\mathbf{s}_t, e)$.

**Attributor term**. Because the candidate set size $m$ varies across time steps, the attributor scores each candidate independently with a scalar salience logit and normalizes via softmax: $P_\eta(Z_t{=}i \mid \mathbf{s}_t, \mathcal{E}_t) \propto \exp g_\eta(\mathbf{s}_t, e_t^i)$, with a learned null logit $g_\eta^\emptyset(\mathbf{s}_t)$ for $i = 0$. It is parameterized by its own LLM backbone, separate from $P_\theta$, with a salience head mapping each encoded $(\mathbf{s}_t, e_t^i)$ pair to its logit. Since $\pi_t$ is fixed, minimizing $\mathcal{L}_{\text{attr}}$ is simply cross-entropy training on the hindsight labels.

**Variational interpretation**. Read as a single objective, Eq. (3) is the negative ELBO of $\log P_{\theta,\eta}(\mathbf{s}_{t+1} \mid \mathbf{s}_t, \mathcal{E}_t)$, with $Q_\phi$ as the variational distribution, $P_\theta$ as the likelihood, and $P_\eta$ as the prior. Unlike a standard VAE, however, the varia-

tional distribution is frozen while the prior is learned, so the bound is loose: its gap, $D_{\mathrm{KL}}(Q_\phi \parallel P_{\theta,\eta}(Z_t \mid \mathbf{s}_t, \mathbf{s}_{t+1}, \mathcal{E}_t))$, never shrinks during training. We therefore regard the objective as *posterior-guided distillation* rather than variational inference; its success rests on the zero-shot calibration of $Q_\phi$, not on tightening a bound.

### 5.3. Inference for Forecasting and Simulation

By decoupling event attribution ($P_\eta$) from transition dynamics ($P_\theta$), a single learned world model naturally supports two distinct downstream tasks. *Forecasting* predicts future market trajectories by marginalizing over the prior attributor's uncertainty, yielding a joint prediction directly comparable against realized data; it thus serves as the natural protocol for validating the world model's overall fidelity. *Simulation* instead bypasses the attributor to condition directly on a specific or hypothetical event, serving as an interventional "what-if" engine—the primary mode in which the world model is deployed for downstream use.

**Forecasting**. Given a candidate event set $\mathcal{E}_t$, the forecasting process marginalizes the social world model over the prior attributor's predicted distribution:

$$\widehat{\mathbf{s}}_{t+1} = \sum_{i=0}^{m} P_\eta(Z_t = i \mid \mathbf{s}_t, \mathcal{E}_t)\, \mathbb{E}_{P_\theta}\big[\mathbf{s}_{t+1} \mid \mathbf{s}_t, e_t^i\big]. \quad (5)$$

Two components contribute to this sum. The null branch ($i=0$) returns the persistence forecast $\mathbf{s}_t$, while each non-null branch ($i \geq 1$) yields $\mathbf{s}_t + \mu_\theta(\mathbf{s}_t, e_t^i)$. Because the attributor weights sum to one, the $\mathbf{s}_t$ baseline factors out cleanly, reducing the forecast to: $\widehat{\mathbf{s}}_{t+1} = \mathbf{s}_t + \sum_{i\geq 1} P_\eta(Z_t{=}i \mid \mathbf{s}_t, \mathcal{E}_t)\, \mu_\theta(\mathbf{s}_t, e_t^i)$. In other words, the forecast equals the persistence baseline plus an attributor-weighted expected move. As the null probability $P_\eta(Z_t{=}0)$ grows, this expected move gracefully shrinks toward zero. Crucially, under assumption (A1), these weights represent uncertainty over which single event acts as the driver, rather than a proportional multi-event decomposition.

**Simulation**. To simulate the impact of a hypothetical event $e_h$, we evaluate the expectation directly: $\widehat{\mathbf{s}}_{t+1} = \mathbb{E}_{P_\theta}[\mathbf{s}_{t+1} \mid \mathbf{s}_t, e_h] = \mathbf{s}_t + \mu_\theta(\mathbf{s}_t, e_h)$, which represents the belief shift implied by that specific shock. Its deviation from persistence, $\mu_\theta(\mathbf{s}_t, e_h)$, corresponds to the abnormal effect of $e_h$ in a classical event-study sense under assumption (A2). Two caveats constrain this interpretation. *Coverage*: The loss $\mathcal{L}_{\mathrm{wm}}$ supervises $P_\theta(\cdot \mid \mathbf{s}_t, e)$ only for events deemed credible causes during training. Consequently, querying an atypical $e_h$ requires out-of-distribution extrapolation, although the parameter-free persistence baseline remains exact. *Identification*: The measured contrast is conditional rather than structural-counterfactual (Pearl, 2009). If assumption (A1) fails and multiple events jointly drive one shift, the learned effect attributed to a single event may be biased upward.

*Table 1*. **Dataset statistics of SWM-Bench.** #Train, #Test, and #Total denote the number of transition triples ($\mathbf{s}_t, \mathcal{E}_t, \mathbf{s}_{t+1}$). #News is the average candidate-set size $|\mathcal{E}_t|$ per triple, and #Markets is the number of prediction markets covered.

| Market | #Train | #Test | #Total | #News | #Markets |
|---|---|---|---|---|---|
| Kalshi | 1841 | 760 | 2601 | 10.6 | 711 |
| Polymarket | 6705 | 3483 | 10 188 | 13.1 | 2537 |

## 6. Experimental Settings

**Data settings**. We construct SWM-Bench from Polymarket and Kalshi historical data (Dec 2022 to Jan 2026) to evaluate news-driven belief forecasting. To avoid saturation by trivial market inertia, we filter the transition triples ($\mathbf{s}_t, \mathcal{E}_t, \mathbf{s}_{t+1}$) for high price volatility and keyword-matched news (filtering and $z$-score subsampling detailed in Appendix B.2). Candidate news $e \in \mathcal{E}_t$ is drawn from a three-day window timestamped strictly before the target observation $t+1$, so no future information is visible. We further guard against leakage by splitting chronologically (training before Nov 1, 2025; testing thereafter) and noting that the Qwen3-8B backbone used by SWM and all fine-tuned baselines was released in April 2025, with a pretraining cutoff strictly preceding the test window. Detailed statistics of Kalshi and Polymarket data are reported in Table 1.

**Baseline settings**. We compare SWM against three categories of methods. (**1**) *Time-series only* baselines forecast solely from the numerical historical window and are trained on SWM-Bench: Autoformer (Wu et al., 2021), DLinear (Zeng et al., 2023), Informer (Zhou et al., 2021), iTransformer (Liu et al., 2024), PatchTST (Nie et al., 2022), TimeMixer (Wang et al., 2024a), and TimesNet (Wu et al., 2022). (**2**) *Prompting-based LLMs* process both the numerical window and all available news to generate forecasts via zero-shot prompting without parameter updates: Qwen3-8B (Yang et al., 2025a), Qwen3.5-397B-A17B (Qwen Team, 2026), GPT-5.5 (OpenAI, 2026), and TimeCAP (Lee et al., 2025a). (**3**) *Fine-tuned LLMs* also use both modalities but are trained directly on SWM-Bench: Time-LLM (Jin et al., 2023), ChatTime (Wang et al., 2025), FNF (Wang et al., 2024c), and LLMForecasting (Halawi et al., 2024). We standardize the backbone where possible: SWM, Time-LLM, FNF, LLMForecasting, and TimeCAP all use Qwen3-8B, while ChatTime requires its own ChatTime-1-7B-Base. All trainable baselines, including SWM, are trained jointly on Polymarket and Kalshi data, but evaluated separately.

**Evaluation settings**. We define the state $\mathbf{s}_t$ with a look-back window of $w = 16$ days and cap the candidate event set at $m = 30$. SWM performs forecasting via joint inference over the prior attributor and world model (Sec. 5.3). We evaluate using four standard metrics: Mean Absolute Scaled Error (MASE), Mean Absolute Error (MAE) to normalize across differing market volatilities, 3-way Directional Accuracy (DA), and Pearson Correlation (Corr) to capture

*Table 2.* **Evaluation results on SWM-Bench.** We report MASE, MAE, directional accuracy (DA), and Pearson correlation (Corr) on Polymarket and Kalshi. For each platform, we evaluate both the full test set (*all*) and the attributed subset (*attr*), which consists of test examples where the posterior attributor ($Q_\phi$ with Qwen3.5-235B) assigns a non-zero probability to at least one candidate news event.

| Method | Polymarket | | | | | | | | Kalshi | | | | | | | |
| | MASE↓ | | MAE↓ | | DA↑ | | Corr↑ | | MASE↓ | | MAE↓ | | DA↑ | | Corr↑ | |
| | All | Attr | All | Attr | All | Attr | All | Attr | All | Attr | All | Attr | All | Attr | All | Attr |
| *Time-series only* | | | | | | | | | | | | | | | | |
| Autoformer | 1.445 | 1.188 | 0.062 | 0.090 | 0.519 | 0.532 | 0.205 | 0.012 | 1.596 | 1.328 | 0.096 | 0.139 | 0.473 | 0.423 | 0.008 | −0.123 |
| DLinear | 1.111 | 0.991 | 0.048 | 0.075 | 0.580 | 0.584 | **0.348** | 0.103 | 1.245 | 1.192 | 0.075 | 0.125 | 0.492 | 0.500 | −0.035 | −0.211 |
| Informer | 1.405 | 1.054 | 0.060 | 0.080 | 0.570 | 0.584 | 0.257 | 0.222 | 1.367 | 1.167 | 0.083 | 0.122 | 0.459 | 0.444 | −0.013 | −0.179 |
| iTransformer | 0.998 | 1.005 | 0.043 | 0.076 | 0.585 | 0.534 | 0.237 | −0.029 | 1.150 | 1.228 | 0.069 | 0.129 | 0.532 | 0.359 | −0.078 | −0.212 |
| TimeMixer | 0.997 | 1.017 | 0.043 | 0.077 | 0.590 | 0.485 | 0.224 | −0.016 | 1.079 | 1.135 | 0.065 | 0.119 | 0.549 | 0.373 | −0.056 | −0.224 |
| TimesNet | 1.015 | 1.035 | 0.044 | 0.078 | 0.577 | 0.474 | 0.264 | −0.018 | 1.133 | 1.187 | 0.068 | 0.124 | 0.543 | 0.310 | −0.023 | −0.127 |
| PatchTST | 0.994 | 1.008 | 0.043 | 0.076 | **0.592** | 0.515 | 0.306 | 0.007 | 1.174 | 1.232 | 0.071 | 0.129 | 0.536 | 0.289 | −0.035 | −0.194 |
| *Time-series + news (Prompting-based LLM)* | | | | | | | | | | | | | | | | |
| TimeCAP | 1.024 | 0.972 | 0.044 | 0.074 | 0.347 | 0.356 | 0.119 | 0.367 | 1.028 | 1.045 | 0.062 | 0.109 | 0.429 | 0.542 | 0.020 | 0.010 |
| Qwen3-8B | 0.997 | 0.935 | 0.043 | 0.071 | 0.375 | 0.419 | 0.197 | 0.377 | 1.034 | 1.039 | 0.062 | 0.109 | 0.454 | 0.528 | 0.041 | −0.012 |
| Qwen3-8B-think | 0.994 | 0.952 | 0.043 | 0.072 | 0.403 | 0.414 | 0.250 | 0.338 | 1.071 | 1.004 | 0.065 | 0.105 | 0.387 | 0.359 | 0.017 | −0.147 |
| Qwen3.5-397B | 1.036 | 0.920 | 0.044 | 0.070 | 0.362 | 0.537 | 0.137 | **0.483** | 1.142 | 1.194 | 0.069 | 0.125 | 0.524 | 0.718 | 0.108 | 0.181 |
| GPT-5.5 | **0.985** | **0.895** | 0.042 | **0.068** | 0.337 | 0.534 | 0.264 | 0.482 | **0.997** | 1.004 | **0.060** | 0.105 | 0.564 | 0.711 | **0.242** | 0.250 |
| *Time-series + news (Fine-tuned LLM)* | | | | | | | | | | | | | | | | |
| Time-LLM | 0.998 | 0.993 | 0.043 | 0.075 | 0.576 | 0.556 | 0.319 | 0.026 | 1.126 | 1.121 | 0.068 | 0.117 | 0.567 | 0.430 | 0.018 | 0.201 |
| ChatTime | 1.186 | 1.091 | 0.051 | 0.083 | 0.560 | 0.512 | 0.220 | 0.009 | 1.458 | 1.288 | 0.088 | 0.135 | 0.514 | 0.493 | −0.028 | −0.247 |
| FNF | 0.986 | 0.961 | **0.042** | 0.073 | 0.377 | 0.392 | 0.218 | 0.226 | 1.126 | 1.161 | 0.068 | 0.121 | 0.430 | 0.401 | −0.020 | −0.174 |
| LLMForecasting | 1.224 | 1.060 | 0.053 | 0.080 | 0.457 | 0.512 | 0.100 | 0.269 | 1.185 | 0.990 | 0.072 | 0.104 | 0.521 | 0.768 | 0.091 | 0.133 |
| SWM (prior) | 1.141 | 0.933 | 0.049 | 0.071 | 0.535 | **0.685** | 0.088 | 0.307 | 1.013 | **0.800** | 0.061 | **0.084** | **0.589** | **0.845** | 0.167 | **0.380** |
| SWM (posterior) | 0.980 | 0.892 | 0.042 | 0.068 | 0.113 | 0.844 | 0.221 | 0.439 | 0.915 | 0.738 | 0.055 | 0.077 | 0.187 | 0.894 | 0.367 | 0.525 |

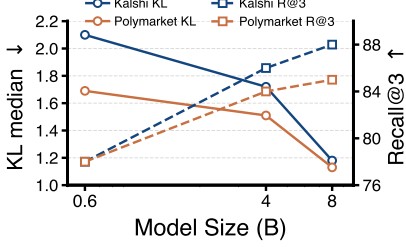

*Figure 3.* **Ablation on the social attributor size.** Qwen3-0.6B/4B/8B are trained with KL and tested on Kalshi and Polymarket. Larger attributors have better performance.

*Figure 4.* **Ablation on the social world model size**. Qwen3-0.6B/4B/8B are trained and tested on Kalshi and Polymarket. Larger world models have better performance.

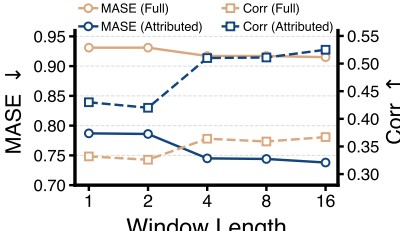

*Figure 5.* **Ablation on the time series window size**. We change the time-series history length from 1/2/4/8/16 in the input. The evaluation is conducted with Kalshi data.

the magnitude tracking of belief changes. Formal metric definitions are provided in Appendix C.4.

# 7. Experimental Results

**Overall performance**. Table 2 demonstrates that SWM is particularly effective at anticipating the direction of news-driven market movements. On the attributed subset, SWM (prior) achieves the highest directional accuracy (DA) among all baselines on both Polymarket (0.685) and Kalshi (0.845), and on Kalshi performs strongly across all four attributed metrics (MASE 0.800, MAE 0.084, Corr 0.380). The takeaway is that our 8B SWM with explicit latent event attribution matches or exceeds frontier LLMs that ingest

the same news, suggesting that our proposed SWM training recipe is the binding factor when event signals are clear.

**Market dynamics difference**. Table 2 also reveals a clear contrast between Kalshi and Polymarket. On Kalshi, SWM surpasses strong baselines, including GPT-5.5 and all time-series models, on nearly every metric, whereas on Polymarket it retains the best directional accuracy but trails massive LLMs in magnitude metrics. This gap likely reflects structural differences. Kalshi questions are tied to fundamental events, producing learnable updates (non-null attribution rate 19%; only 17% of large moves revert). Polymarket potentially hosts heavier algorithmic trading, so more of its moves are endogenous, flow-driven rather than news-driven

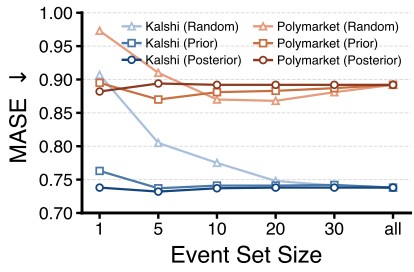

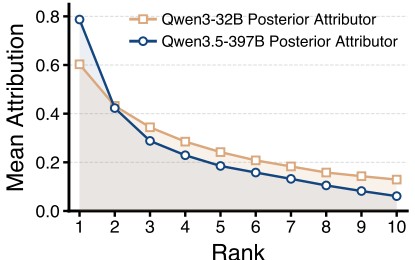

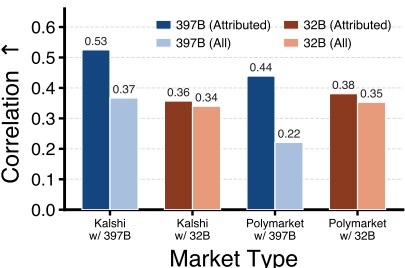

*Figure 6.* **Event set size ablation.** MASE on Kalshi and Polymarket as the event set size $N$ varies, under random, prior-guided, and posterior-guided selection. Posterior saturation supports attribution sparsity.

*Figure 7.* **Posterior attribution distribution.** Mean posterior attribution at ranks 1–10 for Qwen3-32B and Qwen3.5-397B. The 397B posterior is markedly sharper, concentrating mass on the top-1 event.

*Figure 8.* **Effect of posterior attributor on SWM training.** We show correlation on the attributed subset and on the full test set for SWM trained with two posteriors, revealing a precision–coverage trade-off.

(10% attribution rate; 42% revert rate)—precisely the reflexive regime where conditional exogeneity (A2) weakens. The degraded event signal hurts attribution-based training more than purely time-series methods.

**Attribution bottleneck.** Finally, the gray posterior row in Table 2 reports an oracle upper bound, where the hindsight posterior replaces the prior attributor at inference. Performance improves substantially (e.g., Kalshi DA 0.894, Corr 0.525), indicating that the world model $P_\theta$ simulates transitions accurately once the driving event is identified. The critical bottleneck in social forecasting is therefore the prior attributor's ($P_\eta$) ability to isolate and weight the correct causal shock amidst a noisy news cycle.

## 8. Ablation Studies

**Social world model size.** We evaluate SWM with Qwen3 backbones of 0.6B, 4B, and 8B on the attributed subset under posterior-mode inference, isolating the world model from prior-attribution error (Fig. 4). Larger backbones consistently improve prediction: on Kalshi, MASE decreases from 0.884 (0.6B) to 0.738 (8B) while Corr increases from 0.290 to 0.525; on Polymarket, MASE drops from 0.977 to 0.892 and Corr rises from 0.161 to 0.439. The largest marginal gains occur from 0.6B to 4B, suggesting that added capacity primarily strengthens the translation of textual event evidence into numerical belief dynamics.

**Time-series window size.** We vary the historical price window $w \in \{1, 2, 4, 8, 16\}$ supplied to SWM at inference time, under the same posterior-mode setup (Fig. 5). On Kalshi's attributed subset, lengthening the window from $w=1$ to $w=16$ yields modest gains (MASE 0.787 to 0.738; Corr 0.430 to 0.525), with a similar minor trend on the full test set. Since most gains materialize by $w=4$, a brief price history suffices to capture the social momentum relevant for one-step belief prediction.

**Event set size.** We compare three event selection strategies (*random*, *prior-guided*, and *posterior-guided*) across event

pool sizes $m \in \{1, 5, 10, 20, 30, \text{all}\}$ (Fig. 6). The posterior-guided selector matches full-set performance with a single event ($m=1$), and the prior-guided selector converges to the same level by $m=5$, whereas the random selector starts far weaker and requires the full pool to catch up. This accords with *attribution sparsity*: most of the causal signal for a market shift is concentrated in a single relevant event. One exception is Polymarket, where the random selector can surpass the posterior-guided one around $m=10$, suggesting that posterior attribution hindsight is less reliable in Polymarket and may occasionally exclude the truly relevant event.

**Prior attributor model size.** We examine the effect of scaling the prior attributor used to rank candidate news at inference, training Qwen3 attributors at 0.6B, 4B, and 8B. Fig. 3 indicates that larger attributor backbones better approximate the oracle hindsight posteriors. On Kalshi, the median KL divergence to the oracle posterior drops from 2.10 (0.6B) to 1.72 (4B) and 1.18 (8B), while Recall@3—the fraction of transitions where the oracle's top-1 event appears in the attributor's top-3 candidates—rises from 78% to 86% and 88%. A similar pattern holds on Polymarket, with KL dropping from 1.69 (0.6B) to 1.51 (4B) and 1.13 (8B), and Recall@3 rising from 78% to 84% and 85%. Scaling the attributor thus concentrates probability mass on the primary events, providing a more reliable signal for downstream world model training.

**Posterior attributor model size.** The choice of the frozen posterior for training trades off precision against coverage. In Fig. 7, a large Qwen3.5-397B posterior generates a sharp, top-heavy distribution (top-1 mass 0.787) but identifies an external news cause for only 11.9% of transitions, whereas a Qwen3-32B posterior is flatter (top-1 mass 0.603) yet attributes 47.8%. Fig. 8 shows the consequence: the sharper 397B posterior yields higher correlation on the attributed subset (0.525 vs. 0.357 on Kalshi) but lower all-set correlation due to restricted coverage. The optimal choice thus depends on whether the downstream task prioritizes high-confidence causal relations or full-population coverage.

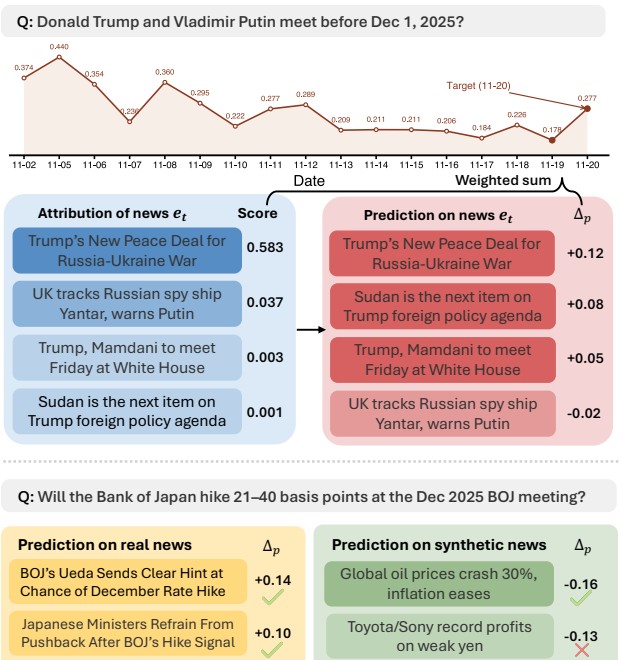

*Figure 9.* **Case studies for forecasting and simulation.** We illustrate the two modes of SWM. (**Top**) *Forecasting:* The prior attributor and the world model jointly generate a prediction by computing an attribution-weighted sum across candidate news events. (**Bottom**) *Simulation:* The world model operates independently to estimate belief shifts conditioned on real or counterfactual events.

## 9. Case Studies

### 9.1. Causality in Posterior Attribution

LLM-based posterior attribution does not formally certify causation. Instead, it acts as a causally *aligned* explanatory filter—surfacing news that best explains a realized move via semantic relevance, temporal precedence, and directional consistency. While highly effective at isolating likely drivers from background noise, it can occasionally latch onto spurious correlations.

**A causally aligned case**. For the contract *"Will the Bank of Japan hike 21–40 bps at the December 2025 meeting?"*, the top-attributed article (attribution score 0.95)—*"BoJ to consider the 'pros and cons' of a rate increase..."*—is published strictly *before* a sharp $0.690 \rightarrow 0.844$ upward adjustment. Here, the attributed news plausibly carries the genuine information shock driving the belief update.

**A spurious case**. The attributor can also latch onto *topical* rather than causal signals. For the contract *"Will Ethereum reach $6,000 by December 31, 2026?"*, the top-attributed article (score 0.75) reports Ethereum trading *flat*—yet it is assigned high responsibility for a $0.320 \rightarrow 0.225$ drop that subsequently reverts. This same article is frequently reused as the "cause" across multiple Ethereum threshold contracts regardless of their actual shift directions. High attribution here reflects mere semantic overlap with the asset, not an underlying causal event.

### 9.2. Forecasting and Simulation Process

Figure 9 demonstrates the utility of decoupling the prior attributor $P_\eta$ from the event-conditioned world model $P_\theta$ across two primary inference modes.

**Forecasting**. When forecasting via joint inference, the model integrates unstructured daily information streams to predict belief trajectory steps. In the Trump–Putin meeting scenario, the baseline price history exhibits a noisy, downward trend that traditional time-series models naively extrapolate. In contrast, SWM applies $P_\eta$ to the candidate pool $\mathcal{E}_t$, assigning the highest prior attribution score (0.583) to the Russia–Ukraine peace deal news, which $P_\theta$ maps to a positive directional shift ($\Delta_p = +0.12$). Moreover, the model successfully captures nuanced negative signals: a news item regarding the UK tracking a Russian spy ship receives a small attribution score but yields a logical negative prediction ($\Delta_p = -0.02$), capturing how escalating geopolitical friction marginally hinders near-term diplomatic summits. The resulting marginal expectation yields an upward correction that closely mirrors the true market jump.

**Simulation**. Conversely, under simulation mode, the model bypasses $P_\eta$ to function as a pure interventional simulator. When directly queried with real news events, $P_\theta$ yields aligned directional estimates (e.g., $\Delta_p = +0.14$ for Ueda's rate-hike signal and $\Delta_p = +0.10$ for the absence of ministerial pushback). This mode also accommodates entirely hypothetical shocks: a synthetic 30% crash in global oil prices correctly triggers a predicted reduction in hike probability ($\Delta_p = -0.16$) due to anticipated deflationary pressures. However, the simulation's error on the weak-yen corporate profit scenario ($\Delta_p = -0.13$) reveals a critical bottleneck. In macroeconomic reality, record corporate profits on a weakening currency exacerbate imported inflation, putting intense pressure on a central bank to *raise* rates ($\Delta_p > 0$). The model instead relies on a surface-level heuristic—assuming that strong corporate performance justifies maintaining the loose monetary status quo. This failure highlights that the fidelity of counterfactual simulation is fundamentally bounded by whether the underlying LLM backbone has successfully internalized multi-step systemic or economic mechanisms.

## 10. Conclusion

We introduced an LLM-based training recipe for building social world models of collective belief evolution, evaluated on our prediction market benchmark SWM-Bench. Enabled by a human-annotation-free, posterior-guided training paradigm that isolates latent causal drivers, SWM achieves state-of-the-art performance on Kalshi and competitive results on Polymarket. We hope this work motivates future research on parametric social world models.

## Impact Statement

This work introduces the Social World Model (SWM) and SWM-Bench, a framework and dataset designed to predict how collective beliefs evolve in response to global events. By grounding social dynamics in prediction markets and leveraging the reasoning capabilities of LLMs, this framework offers a scalable toolkit for quantitative digital sociology. The primary positive impact of SWM lies in its capacity to assist policymakers, non-governmental organizations (NGOs), and economists in anticipating public reactions to critical developments—such as public health crises, macroeconomic shocks, or policy interventions. This predictive capability enables more proactive, informed, and resilient governance that better aligns with societal needs.

However, modeling shifts in social consensus carries inherent ethical risks. If deployed maliciously, a social world model could be weaponized to optimize targeted disinformation, engineer persuasive social engineering campaigns, or manipulate public opinion by identifying the specific informational stimuli required to trigger a desired belief shift. Additionally, because SWM relies on prediction markets as a proxy for public sentiment, deploying such systems in live environments risks creating algorithmic feedback loops, where the model's public forecasts inadvertently amplify market volatility or artificially distort the very consensus they intend to measure.

To mitigate these concerns, we advocate for the transparent and defensive deployment of social world models. Rather than optimizing for public intervention, these tools are best suited to detect unnatural, abrupt, or inorganic shifts in collective belief, thereby helping researchers identify and counter coordinated influence operations. Ultimately, while SWM provides a powerful lens into the collective human psyche, its application must be governed by rigorous ethical oversight to ensure it serves to protect open information ecosystems rather than exploit them.

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

## A. The Use of Large Language Models (LLMs)

We used ChatGPT as a writing assistant for parts of the paper and GitHub Copilot/Claude Code/Codex to accelerate coding. All AI-assisted writing and code were manually checked and revised; no content in the paper is fully AI-generated.

## B. Asset Details

### B.1. Code and Data Open-source

We release our complete dataset under the Creative Commons Attribution-NonCommercial-NoDerivatives 4.0 International (CC BY-NC-ND 4.0) License and the code under the MIT License. Since our dataset is derived from publicly accessible information on Polymarket and Kalshi, we ensure compliance with both platforms' Terms of Service by avoiding any restricted activities and including clear attribution in our release materials. This licensing choice supports open research while safeguarding both ethical reuse and legal compliance.

### B.2. Dataset Details

We outline below the technical details of our dataset collection, which includes both news data and social belief data from Polymarket and Kalshi.

**News data collection**. We use the gnews API service (`https://gnews.io`) to collect daily news from December 2022 through January 2026, obtaining more than 10 significant news headlines and descriptions per day.

**Candidate event set construction**. For each transition $(\mathbf{s}_t, \mathcal{E}_t, \mathbf{s}_{t+1})$, we build the candidate set $\mathcal{E}_t$ by querying the GNews API with keywords extracted from the market question, restricting results to a three-day window preceding the target observation $t+1$ (so all candidates are timestamped strictly before $t+1$), and capping the set at most 100 candidates but we randomly sample 30 of them for training and inference.

**Polymarket and Kalshi data collection**. To collect real-world belief data, we use the publicly accessible APIs of Polymarket (`https://gamma-api.polymarket.com`) and Kalshi (`https://kalshi.com`). We gather both metadata (e.g., market title, status, tags) and historical time-series data for each market.

**Z-score sub-sampling**. Raw prediction-market series are overwhelmingly dominated by no-change points—on most days a contract's price barely moves—so a benchmark built from raw samples would mostly reward copying the last price and would say little about a model's ability to read news. Since our goal is to evaluate *news-driven* forecasting, we sub-sample transitions by the statistical significance of their move rather than keeping the flat majority. For each candidate point we compute a $z$-score of the latest price change relative to the contract's recent volatility,

$$z = \frac{|p_t - p_{t-1}| - \mu_{\mathrm{w}}}{\sigma_{\mathrm{w}}}, \tag{6}$$

where $\mu_{\mathrm{w}}$ and $\sigma_{\mathrm{w}}$ are the mean and standard deviation of $\{|p_i - p_{i-1}|\}$ over a trailing 30-step window (requiring at least 10 prior steps). We then sub-sample low-$z$ (near-flat) points and retain statistically significant moves, yielding a benchmark concentrated on the regime where news actually drives price changes.

### B.3. Model License

We include the licenses for all models used during training, inference, and data collection:
**Qwen3-0.6B** License: Apache 2.0
**Qwen3-4B** License: Apache 2.0
**Qwen3-8B** License: Apache 2.0
**Qwen3-32B** License: Apache 2.0
**Qwen3.5-397B-A17B** License: Apache 2.0
**GPT-5.5** License: Proprietary (OpenAI)
We used OpenAI's GPT-5.5, a proprietary large language model accessible via API (`https://openai.com`). Usage complies with OpenAI's Terms of Use (`https://openai.com/policies/terms-of-use`).

*Table 3.* **Training hyperparameters.** Both components fully fine-tune all parameters (no LoRA) in bf16 under FSDP full-shard with activation/gradient checkpointing; the world model's regression head is kept in fp32.

| | Backbone | LR | Batch Size | Epochs | Warmup | Max length | Max event size |
|---|---|---|---|---|---|---|---|
| Prior attributor $P_\eta$ | Qwen3-0.6B/4B/8B | $10^{-5}$ | 28 (eff.) | 1 | — | 1024 | 30 |
| World model $P_\theta$ | Qwen3-0.6B/4B/8B | $2 \times 10^{-5}$ (5× head) | 4 | 6 | 30 | 1024 | 30 |

# C. Experimental Details

## C.1. Compute Resources

All training experiments are conducted on 8 A100 80GB GPUs; inference runs on a single A100 80GB GPU.

## C.2. Training Details

Training follows the pipeline of Figure 2: the frozen posterior attributor $Q_\phi$ first labels all training transitions offline, after which the prior attributor $P_\eta$ and the world model $P_\theta$ are trained independently on these labels. Both trainable components use Qwen3 backbones, chosen for strong performance at small scales that remain deployable for real-time use on Polymarket and Kalshi; the posterior attributor instead requires a state-of-the-art model to produce reliable pseudo-labels. Prompt design plays a minor role for the trained components, as fine-tuning removes most reliance on prompt engineering; all templates are in Appendix D and hyperparameters in Table 3.

**Posterior attributor ($Q_\phi$) hindsight.** The posterior attributor uses a structured prompt passed to the Qwen3.5-397B-A17B model. For each candidate news $e_t^i$ it returns an independent Bernoulli responsibility score $s_i \in [0, 1]$—the probability that $e_t^i$ drives the realized move. To turn these per-news scores into attribution weights, we map each to its odds and renormalize against a fixed no-news pseudo-event (score 0.5, i.e. odds $\rho_0 = 1$):

$$o_i = \frac{s_i + \epsilon}{1 - s_i + \epsilon}, \qquad \pi_i = \frac{o_i}{\rho_0 + \sum_j o_j}, \tag{7}$$

with $\epsilon = 10^{-3}$ for numerical stability. When every $s_i$ is small, the mass falls on the null category (which predicts no change), whereas a single confident news item ($s_i \to 1$) takes over; directly normalizing the $s_i$ to sum to one would instead force weak news to absorb full weight and spuriously predict large moves on transitions with no real news signal.

Formally, this is the Bradley–Terry/Luce construction (Bradley & Terry, 1952; Luce et al., 1959): interpreting each $s_i$ as a pairwise comparison of candidate $i$ against the null hypothesis, $s_i = w_i/(w_i + w_0)$ with the null strength anchored at $w_0 = 1$ (i.e., $s = 0.5$), the odds $o_i$ recover the latent strengths $w_i$, and the normalization yields the unique categorical distribution consistent with the elicited pairwise probabilities under Luce's choice axiom; equivalently, $\log o_i$ are multinomial-logit scores with the null as the base category (Becg & Gray, 1984). This interpretation is faithful to the elicitation process, since the LLM scores each candidate independently—each $s_i$ is precisely a binary judgment of "this news vs. no news."

**Prior attributor ($P_\eta$) training.** The prior attributor amortizes the posterior attributor into a single forward pass, so attribution runs online without querying the 397B teacher. It is a Qwen3 backbone (0.6B/4B/8B) with a scalar regression head: each (market context, candidate news $e_t^i$) pair is encoded *independently* to a logit $z_i$, a "no relevant news" prompt yields a null logit $z_0$, and a per-transition softmax over the $m$ candidates and the null category gives the predicted weights $q$ (with a shared temperature $T = 0.5$ at training and inference, an implementation detail absorbed into the salience logits $g_\eta$ of the main text).

We distill the posterior weights $\pi$ as a soft target by minimizing the per-transition forward KL $\mathrm{KL}(\pi \| q)$. Because $\pi$ assigns explicit mass $\pi_0$ to the no-news category, the objective supplies gradient to $z_0$ directly, teaching the prior to abstain on transitions with no causal news rather than concentrating weight on a weak candidate. Since $\sim 75\%$ of training transitions carry no attributed news, we subsample null transitions to an $\approx 1{:}1$ ratio, and train for a single epoch: held-out test KL is minimized at $\approx 1$ epoch and degrades thereafter as the model overfits the teacher's training markets.

**World model ($P_\theta$) training.** For each transition, the model encodes each $(\mathbf{s}_t, e_t^i)$ pair and regresses the predicted shift $\mu_\theta(\mathbf{s}_t, e_t^i)$ toward the realized $\Delta_t$, weighted by the renormalized posterior weights $\bar{\pi}_t^i$, implementing the $\mathcal{L}_{\mathrm{wm}}$ of Sec. 5.2.

## C.3. Inference Details

At inference, the prior attributor and the world model jointly form the forecast of Sec. 5.3: $P_\eta$ scores each candidate news item (and the null category) to produce attribution weights, $P_\theta$ predicts the event-conditioned shift $\mu_\theta(\mathbf{s}_t, e_t^i)$ for each candidate, and the final prediction is the attribution-weighted sum $\widehat{\mathbf{s}}_{t+1} = \mathbf{s}_t + \sum_{i \geq 1} P_\eta(Z_t{=}i \mid \mathbf{s}_t, \mathcal{E}_t)\,\mu_\theta(\mathbf{s}_t, e_t^i)$. In posterior (oracle) mode, the hindsight weights $\pi_t$ replace $P_\eta$.

## C.4. Evaluation Details

Let the test set contain $N$ examples. For example $i$, let $p_i$ be the last observed price (at time $t$), $y_i$ the realized next price (at $t{+}1$), and $\hat{y}_i$ the predicted price, with realized and predicted belief changes $\Delta_i = y_i - p_i$ and $\hat{\Delta}_i = \hat{y}_i - p_i$.

**Mean Absolute Error (MAE)**. The average magnitude of the prediction error:

$$\text{MAE} = \frac{1}{N} \sum_{i=1}^{N} |\hat{y}_i - y_i|. \tag{8}$$

**Mean Absolute Scaled Error (MASE)**. MAE normalized by the error of the *persistence* (no-change) baseline $\hat{y}_i = p_i$, which scales away cross-market differences in volatility; MASE $< 1$ means the model beats copying the last price:

$$\text{MASE} = \frac{\sum_i |\hat{\Delta}_i - \Delta_i|}{\sum_i |\Delta_i|}. \tag{9}$$

**Three-way Directional Accuracy (DA)**. We classify each change as *up*, *down*, or *no-change* via $\text{sgn}_\delta(x) \in \{+1, -1, 0\}$ with a small dead band $\delta = 10^{-6}$ ($+1$ if $x > \delta$, $-1$ if $x < -\delta$, $0$ otherwise), and measure agreement over the *moved* subset $\mathcal{M} = \{i : |\Delta_i| > \delta\}$:

$$\text{DA} = \frac{1}{|\mathcal{M}|} \sum_{i \in \mathcal{M}} \mathbb{1}\Big[\text{sgn}_\delta(\hat{\Delta}_i) = \text{sgn}_\delta(\Delta_i)\Big]. \tag{10}$$

The three-way scheme penalizes predicting "no change" on a market that genuinely moved.

**Pearson Correlation (Corr)**. The Pearson correlation between predicted and realized belief changes over all $N$ examples, capturing how well the model tracks the magnitude and sign of moves:

$$\text{Corr} = \frac{\sum_i (\hat{\Delta}_i - \bar{\hat{\Delta}})(\Delta_i - \bar{\Delta})}{\sqrt{\sum_i (\hat{\Delta}_i - \bar{\hat{\Delta}})^2}\,\sqrt{\sum_i (\Delta_i - \bar{\Delta})^2}}, \tag{11}$$

where $\bar{\hat{\Delta}}$ and $\bar{\Delta}$ are the respective means.

**Scope**. MAE, MASE, and Corr are computed over the evaluated subset (*all* or *attributed*); DA is computed over the moved subset $\mathcal{M}$ within that subset. MAE and MASE are lower-is-better; DA and Corr are higher-is-better.

# D. Detailed Prompts

We provide full prompts mentioned in Appendix. C.2, including prompts for SWM, posterior attributor, and prior attributor.

```
Prompt for SWM

You are given a prediction market, its recent price history, and a relevant news
article. Your task is to predict the market probability on the target date.

[Prediction Market]
Event: {record.question}
Description: {record.description}

[Recent Price History]
```

```
{date_1}: {price_1}
{date_2}: {price_2}
...
{date_N}: {price_N}

[News Article]
Title: {news.title}
Content: {news.description}

[Forecast Target]
Predict the probability on {target_date}.

[Output Format]
Return only a number between 0 and 1.
Do not include any explanation.
```

## Prompt for posterior attributor

```
You are searching for the CAUSAL FACTOR behind a specific price change in a prediction market. Your
    task: decide whether THIS news article is a causal driver of THAT specific change.

Note: cases where |z| < 1 (no significant deviation from market's typical volatility) are filtered
    out upstream and never reach you -- every case you see has |z| >= 1 and warrants causal analysis.

=== 6 EXAMPLES (study before scoring) ===
...
=== NOW SCORE THIS CASE ===

MARKET QUESTION: {question}
{description_block}

PRICE TRAJECTORY (17-day window leading up to and including the change):
{history_str}

THE CHANGE TO EXPLAIN:
  {date_before} -> {date_after}:  {price_before:.3f} -> {price_after:.3f}  ({arrow} {abs_price_change
    :+.3f}, {direction})
  |Delta p| = {abs_price_change:.3f}

NEWS ARTICLE (CANDIDATE CAUSE):
  Published: {pub_date}
  Title: {title}
  Content: {desc}

REASONING CHECKS (apply each):
  1. DIRECTIONAL FIT: Would a rational trader, after reading this, push the
     price in the OBSERVED direction ({direction})?
     If OPPOSITE -> score 0 (like Examples 2 and 3).

  2. COUNTERFACTUAL: If this specific news did NOT exist, would the price still
     have moved by approximately {abs_price_change:.3f} on this day?
     If YES -> low score; this news is not load-bearing.

  3. MECHANISM: Does this news contain new information that would shift trader
     beliefs about the market question? The mechanism can be DIRECT (Example 1)
     or MULTI-HOP / INDIRECT (Example 5) -- both are valid causal mechanisms.
     If only vague topical overlap with NO mechanism -> low score (Example 4).

SCORING SCALE:
  90-100  Singular cause (like Example 1).
```

```
 70-89    Strong contributor with clear causal mechanism in observed direction.
 40-69    Plausible contributor, but other factors likely also at play.
 10-39    Topically related but no specific causal mechanism (like Example 4).
 0-9      Unrelated, OR wrong direction (Examples 2, 3), OR no information.

CRITICAL: Most candidate news in a pool deserves a LOW score (0-30). It is
correct and expected to output 0 when you find no causal link. Do NOT feel
pressured to find a connection -- only score high when the causal chain is clear
(matching direction + counterfactual fails + specific mechanism).

Output: a single integer 0-100. No explanation.
```

## Prompt for Prior Attributor

```
You are given a news article and a prediction market. Your task is to estimate
whether the news article is causally related to a subsequent price change in the
prediction market.

[News Article]
Date: {news.published_at}
Title: {news.title}
Content: {news.description}

[Prediction Market]
Question: {record.question}
Description: {record.description}

[Forecast Target]
Predicting for: {target_date}

[Recent Price History]
{date_1}: {price_1}
{date_2}: {price_2}
...
{date_N}: {price_N}

[Task]
Does this news article have a causal relationship with the price change of this
prediction market?

Assign a higher score only if the news could plausibly and directly cause the
market price to move. News that is merely topically related, but would not change
traders' beliefs about the market outcome, should receive a low score.

[Scoring Guidance]
- High score: the news provides new information that could directly shift the
  market probability.
- Medium score: the news is plausibly relevant, but the causal link is indirect
  or uncertain.
- Low score: the news is only topically related or does not provide information
  that would affect the market price.
- Very low score: the news is unrelated to the market question.

[Output Format]
Return a single integer score from 0 to 100.
Do not include any explanation.
```

