# OpenReview forum: "Building Social World Model with Large Language Models"
_ICML.cc/2026/Conference — ICML 2026 regular_

### Official Review · Reviewer_6cjR · 2026-03-11

**Soundness:** 2
**Presentation:** 3
**Significance:** 3
**Originality:** 2
**Overall Recommendation:** 3
**Confidence:** 4

**Summary:**

The paper proposes a general framework SWM to capture the evolution of social beliefs.
SWM learns state transition functions for social beliefs by mining temporal patterns in social data. SWM-bench is introduced to evaluate the performance of time-series prediction.

**Compliance With Llm Reviewing Policy:**

Affirmed.

**Key Questions For Authors:**

1.Transition function only consider one-step as formualtion (1), however, actual event might have long term dependence.
2.What is social beliefs? Are there further restrictions on the "proposition" within the definition? The concept appears rather vague. Based on the given instances, it seems that any problem could potentially qualify as q.
3.The social belief appears similar to the perspective of events analysis, and there are already many methods for perspective analysis, which can also be accomplished using LLMs.
4.Many financial forecasts are also event-driven, not solely reliant on time series. Hence, I think the explaination of SWM, it should not only be compared with standard time series predictions.
5.The method actually requires to use market price data as a marker, and the selection of market should done by human. The expression "without the need for explicit human annotations" is not suitable.

**Limitations:**

yes

**Strengths And Weaknesses:**

Weaknesses:
-Social models encompass a wide range of issues, however, the paper just focus on market-related news and price data.
-The comparative experiments are not sufficiently comprehensive, especially for those methods based on event-driven market prediction.

---

> ### Author Rebuttal · Authors · 2026-03-31
>
> **[Baseline clarification]** Our experiments already include both standard time-series baselines (Autoformer, DLinear) and event-aware LLM-based baselines (Time-LLM, GPT-5.2) that incorporate textual event information (Lines 286–301). The evaluation is therefore not limited to pure time-series forecasting.
>
> **[Additional time-series baselines]** We add ChatTime-Base [1], ChatTime-Chat [1], and From News to Forecast [2] as additional event-driven time series baselines. These methods jointly leverage time series data and event/news data. We also add Informer and Reformer as additional time-series baselines.
>
>
> | Method | Polymarket MAE ↓ | Polymarket RMSE ↓ |
> |---|---|---|
> | Autoformer | 0.1476 | 0.1182 |
> | DLinear | 0.1100 | 0.0815 |
> |Informer|0.0703|0.0979|
> |Reformer|0.0687|0.0959|
> | TimeLLM | 0.0800 | 0.0612 |
> |ChatTime-Base | 0.1000 | 0.1413|
> |ChatTime-Chat|0.0883|0.1275|
>  |From news to forcast| 0.0634 | 0.0782|
> | SWM | 0.0649 | 0.0560 |
>
> **[Additional frontier model baselines]** Besides GPT-5.2, we also include frontier models like Deepseek-V3 and R1. But these models have lower performance compared with GPT-5.2.
>
> | Model | MAE ↓ | RMSE ↓ |
> |---|---|---|
> | SWM | 0.0649 | 0.0560 |
> | GPT-5.2 | 0.0587 | 0.0522 |
> | DeepSeek-V3 | 0.1069 | 0.1615 |
> | DeepSeek-R1 | 0.1752 | 0.2332 |
>
>
> **[Long-term dependency in transition funcations]** The one-step formulation is a deliberate design choice matched to the nature of our data. (1) Our data contains both sudden jumps and gradual shifts. Roughly half of our training data consists of sharp breakpoint jumps driven by discrete news events, while the other half captures smoother belief evolution. For the sudden-change portion, the causal signal is concentrated at the one-step horizon; for the gradual portion, one-step prediction captures the incremental drift without forcing the model to disentangle multiple compounding causes over longer horizons. In both cases, one-step provides the cleanest supervision for isolating event effects. (2) One-step composes into multi-step via autoregressive rollout. A well-learned transition function can be applied iteratively: $s_t \to s_{t+1} \to \cdots \to s_{t+k}$, with fresh event retrieval at each step—preserving clean event grounding while capturing longer-range dynamics through composition. Evaluation of multi-step rollout is left for future work.
>
> **[Proposition of social beliefs]** Social beliefs can in principle be defined over any proposition that a large population holds opinions about—not restricted to market questions. However, not every social belief is worth modeling: many are too trivial, too stable, or lack observable data. We focus on the subset that is (1) important to a large population, (2) highly uncertain, and (3) dynamically influenced by public information. Prediction markets are a natural testbed—not because they define social belief, but because they surface questions people care about, disagree on, and continuously update, while market prices provide an observable behavioral signal of evolving collective beliefs.
>
>
> **[Difference from perspective/event analysis]** SWM differs fundamentally from both tasks. Perspective analysis [1] classifies the stance expressed in a single text. Classical event studies [2] measure how a known event affects asset prices over a window. SWM models neither—it models the temporal evolution of a latent collective belief state: given $s_t$, predict $s_{t+1}$ and attribute which candidate event drove the shift. Event analysis is a necessary sub-ability within SWM, but the modeling target is belief dynamics under external events, not static opinions or price responses to known shocks.
>
>
>
> **[Explanation for no need for explicit human annotation]** We clarify the distinction between data source and method annotation. Market prices are indeed produced by human trading and reflect collective human beliefs—this is precisely why they serve as a high-quality dataset for social belief dynamics. The markets themselves are also naturally selected by human interest. However, this human involvement is inherent in the data source, not an annotation requirement of our method. Our claim is that given such data, SWM requires no additional human labels for training—e.g., no one manually annotates which event caused a particular belief shift. The posterior attribution and world model training are both learned from market price signals alone.
>
> [1] Saif Mohammad et al. SemEval-2016 Task 6: Detecting Stance in Tweets. ACL 2016
>
> [2] Xiao Ding et al. Deep Learning for Event-Driven Stock Prediction. IJCAI 2015

---

### Official Review · Reviewer_q1pu · 2026-03-12

**Soundness:** 3
**Presentation:** 3
**Significance:** 3
**Originality:** 3
**Overall Recommendation:** 5
**Confidence:** 4

**Summary:**

This paper proposes the Social World Model (SWM), a versatile framework designed to leverage large language models (LLMs) to simulate the dynamic response of social beliefs to external events. Existing methods often lack high-quality annotated data when capturing the temporal evolution of social beliefs and struggle to model dynamic event perturbations. To address this, the authors make three primary contributions: First, they construct the SWM-Bench, aggregating over 10k high-fidelity social belief prediction time series from Polymarket and Kalshi. Second, they design a posterior-guided variational training paradigm that attributes events via a latent variable Zt, utilizing a posterior attributor Qφ to guide a prior attributor Pη in generating pseudo-labels. Finally, they formalize the transition process of social states perturbed by events. Experiments on real-world market datasets demonstrate that SWM outperforms time-series baselines like TimeLLM in prediction accuracy and exhibits strong generalization capabilities in specific domains.

**Compliance With Llm Reviewing Policy:**

Affirmed.

**Key Questions For Authors:**

1. Regarding Attribution Validity: Have the authors conducted any qualitative manual spot-checks on the key events output by the model? How can it be proven that the events filtered by the model are the "causes" driving belief changes, rather than merely "effects" co-occurring with high-volatility windows?

2. Regarding Candidate Filtering: Could you elaborate on the specific filtering algorithms or heuristic rules used to transition from raw Google News search results to the final candidate event set E_t? Does this process introduce selection bias?

3. Regarding Computational Cost: Compared to mainstream time-series baseline models, how much does the inference latency of SWM increase? While pursuing prediction accuracy, is this computational trade-off still viable for high-frequency or real-time application scenarios?

4. Regarding Demographic Bias: How do the authors view the demographic representation bias inherent in the training data sources (prediction market platforms)? What recommendations exist to mitigate this bias when extending the model to other social issues?

**Limitations:**

Yes.

**Strengths And Weaknesses:**

Strengths

1. Novel Problem Formulation: Combining the semantic reasoning capabilities of LLMs with the belief dynamics of quantitative markets is a highly promising interdisciplinary direction. Modeling social events as exogenous perturbations in time series provides a computationally friendly paradigm for causal discovery in social dynamics.

2. Solid Dataset Contribution: SWM-Bench is a major highlight of this work. The field of social dynamics simulation generally lacks structured time-series data; utilizing prediction markets as a high-fidelity proxy for social beliefs effectively fills this critical gap.

3. Inspiring Methodological Design: Introducing a posterior-guided variational inference mechanism in a weakly supervised setting without explicit attribution labels is highly ingenious. The idea of distilling knowledge from Qφ to Pη logically closes the loop on the unsupervised attribution challenge.

4. Comprehensive Empirical Evaluation: The experimental design is rigorous, with SWM outperforming baselines comprehensively on core metrics like MAE and RMSE. The ablation studies  clearly disentangle the impact of window size and candidate set scale on model performance, validating the effectiveness of each module.

Weaknesses

1. Questionable Rigor in Causal Interpretation: Although the paper emphasizes "interpretability" and "causal reasoning," the core evaluation still relies on prediction accuracy (MAE/RMSE). Does the "important news event" assigned a high weight by the model represent a true causal driver? There is currently a lack of qualitative expert evaluation or user studies to verify this, running the risk that the model is merely fitting statistical correlations.

2. Lack of Detail in Candidate Event Set (Et) Construction: The paper mentions retrieving candidate events via Google News but does not detail core preprocessing steps such as deduplication or relevance filtering thresholds. If the initial event pool contains significant noise, does the posterior attributor have sufficient robustness to extract true signals?

3. Insufficient Discussion on Computational Overhead and Real-time Feasibility: The training involves variational inference and marginalization over candidate events. When the candidate set size $m$ is large, the computational burden inevitably increases. The paper fails to provide a quantitative comparison of training costs and inference latency against standard time-series models like Autoformer.

4. Potential Dataset Bias: The user demographics of Polymarket and Kalshi exhibit distinct characteristics (e.g., skewing towards tech-oriented populations). The "social dynamics" learned by the model may disproportionately reflect the belief biases of specific demographic circles rather than a universal societal consensus. This limits the model's generalization capabilities across broader demographic contexts.

---

> ### Author Rebuttal · Authors · 2026-03-31
>
> **[Causal validity of attribution]**  As discussed in Lines 404–418, our method cannot guarantee causality in the attribution data. However, this does not undermine the contribution: (1) The goal of SWM is to model event-conditioned social belief dynamics, where public events serve as observable conditioning signals—not to make strict causal claims. (2) SWM's gains over pure time-series baselines (e.g., DLinear, Autoformer) suggest the model captures more than statistical price correlations, especially on large price jumps where temporal extrapolation is insufficient.
>
> **[Qualitative expert evaluation of attribution]** For the market "Will tariffs on copper come into effect before Jan 2026?", the price rises from 0.43 to 0.77 (+34%) between Oct 31 and Nov 1, 2025. The attributor ranks candidates as follows. Rank 1 is directly relevant to the tariff question. Rank 2 is not a direct policy signal, but reflects copper supply disruption—plausibly connected to belief updates on copper-related policy. Rank 3 is plainly irrelevant and correctly scored 0. We additionally conducted human evaluation on a subset of posterior attributions and found that while some high-scored items are false positives (i.e., assigned high attribution but not truly causal), dominant factors are generally ranked correctly.
>
> | Rank | News | Score |
> |---|---|---|
> | 1 | Senate Rebels Against Trump: Lawmakers Vote to Strip President of Power to Slap Tariffs | 1.0 |
> | 2 | African Copper Exports to China Disrupted Amid Tanzania Unrest | 0.9 |
> | 3 | Chiefs Trade Proposal Secures Vikings RB Aaron Jones | 0.0 |
>
>
>
> **[Candidate event set selection]**  Constructed in three steps: (1) GPT-4o-mini extracts 2–3 market-specific keywords. (2) We query Google News (GNews API) with these keywords in a 2-day window and retain the top 100 items, represented by title + 300-char description. We do not perform explicit deduplication—near-duplicate articles receive similar scores, effectively concentrating mass on the same underlying event without distorting rankings. Keyword-based retrieval may miss obliquely relevant events, but the null event design mitigates this by absorbing mass when no candidate is confidently causal.
>
> **[Robustness of posterior attributor]** Our pipeline handles noisy candidates at multiple levels (see also [reliability of posterior attribution] and [null event ablation] in Reviewer WDi2): (1) *Constrained retrieval*. Candidates are filtered by market-specific keywords and a 2-day window, reducing unrelated news before attribution begins. (2) *Sparse attribution*. The posterior concentrates mass on 1–3 dominant events and suppresses the rest—noisy candidates receive near-zero scores and contribute little to the final expectation. (3) *Null event*. When no candidate is confidently causal, probability goes to a dedicated null event rather than being spread over irrelevant news. (4) Empirical validation. Figure 4 (Lines 385–394) shows that adding more candidate events at inference does not degrade prediction, confirming the attributor is not sensitive to moderate retrieval noise.
>
>
> **[Demographic bias and mitigating method]**  Kalshi and Polymarket are the two largest prediction market platforms; using both reduces single-source demographic bias. When extending SWM to other domains, mitigation should operate at data-source and modeling levels: (1) incorporating heterogeneous belief sources (polling, surveys, social media), (2) reweighting based on known participation skews, and (3) modeling source identity to separate shared trends from source-specific artifacts.
>
> **[Training cost]** Training overhead is moderate because posterior attribution is sparse: for each market-timestep pair, only 2–10 candidate events receive non-negligible posterior mass (Lines 243–248). The expectation in Eq. 3 (Lines 269–271) therefore marginalizes over a small effective set, keeping computation manageable even when the full candidate space is large. It is still much higher than the training cost of time-series models like Autoformer.
>
> **[Inference latency]** SWM targets mid/low-frequency, news-driven belief updates where second-level latency is acceptable—it is not designed for high-frequency or millisecond-level forecasting. Compared with single-model baselines like TimeLLM, SWM introduces additional cost from (1) one batch pass of the attributor over candidate events and (2) one batch pass of the forecaster conditioned on non-zero attributed events, followed by weighted aggregation. This results in roughly 2–4× higher inference cost; if the candidate event space exceeds a single batch, inference time scales linearly with the number of batches.
>
> | Method | Inference Components |  GPU time per-DP (avg on A6000 Pro) |
> |---|---|---:|
> | TimeLLM | Single forecasting model | 158ms |
> | SWM | Attributor model (batch inference) + forecasting model (batch inference) | 395ms |

---

> > ### Author Rebuttal · Reviewer_q1pu · 2026-04-02
> >
> > Fully resolved

---

### Official Review · Reviewer_WDi2 · 2026-03-17

**Soundness:** 3
**Presentation:** 3
**Significance:** 3
**Originality:** 4
**Overall Recommendation:** 6
**Confidence:** 4

**Summary:**

The paper presents an approach for modelling social beliefs using a novel world model framework that uses "free markets" as the signal for social sentiment and introduces a benchmark, SWM-Bench, that consists of prediction market data (Kalshi and Polymarket). The model seeks to mix traditional time-series modelling with language data by modelling social beliefs and social events as discrete trajectories of textual event descriptions and prediction market scores or time steps, respectively. The application of LLMs to this setting helps to bridge the gap between past events and state changes by computing semantic alignment.

Experiments are conducted on SWM-Bench. The SWM is compared against both traditional time series models and time series & news models, which includes frontier foundation models and specialized time series foundation models. The experiments report strong performance by SWM with a small LLM backbone (Qwen3-4B), achieving the highest specialized performance and achieving comparable performance to GPT-5.2.

**Compliance With Llm Reviewing Policy:**

Affirmed.

**Final Justification:**

My primary concerns were with the quality of the posterior attributions, ablation coverage, and baseline coverage. The authors provided experimental results that address these concerns. The posterior attribution comparison between 4o-mini and 5.4 and cost reporting provides needed context for why their choice of model is suitable. The authors also provide additional baselines and ablation studies, as requested. In light of this, I believe my concerns have been addressed, and I will raise my score to a 6 as I believe the paper is of high quality.

**Key Questions For Authors:**

Could you please clarify how many posterior attributions were performed by GPT-4o? And the cost?

How critical to the pipeline are the quality of the posterior attributions, and have you attempted any stronger/weaker models?

**Limitations:**

The authors mention that models predicting social beliefs can be used maliciously and advocate for 'defensive' use.

**Strengths And Weaknesses:**

Strengths:

The paper introduces a very interesting framing around building models that predict social beliefs, taking advantage of the recent influx of prediction market data.

The authors clearly outline of the structure of the model and provide thorough descriptions of the problem framing and training decisions

The experimental results show consistent strong performance when compared with the baselines despite using a smaller base model.

Weaknesses:
Because the authors are outlining a complex framework, the implementation of some aspects of the SWM framework are not immediately clear. For instance, the posterior attribution appears to be done via LLM-as-a-judge, which is not immediately clear and impacts the potential reliability of the posterior generation.

The results would be stronger with additional time series baselines (for additional insight into the impact of event modelling) and additional frontier model baselines to provide breadth to the impact of general model strength.

Similar to the prior point, the ablation study focuses mainly on hyperparameters, but it would also provide valuable insight into the impact of framework decisions to ablate components of the pipeline, for instance, removing the null event, experimenting with alternate generations of textual descriptions of events, or different methods of posterior attribution.

Typos:
- "relavent"->relevant line 263 column 2

---

> ### Author Rebuttal · Authors · 2026-03-31
>
> **[Collection of posterior attribution]** Posterior attributions are collected by prompting an LLM with each $(S_t, E_t, S_{t+1})$ triplet and asking it to score which news events most plausibly caused the observed change (prompt in Lines 715–726).
> (1) Scale: We collect attributions for every triplet—3,127 (Kalshi) and 10,503 (Polymarket). The total attributed news are 232,647 (Kalshi) and 765,587 (Polymarket). (2) Cost: We batch 10 news items per prompt. Using GPT-4o-mini, total cost was approximately \$100–\$200.
>
> **[Reliability of posterior attribution]** We will make the LLM-as-judge attribution procedure more explicit in the revision. (1) *Ground truth is unobservable and hard to collect*, but the target is sparse. Sudden price changes are typically driven by 1–5 dominant factors, so what matters is whether the estimate recovers the top causal factors, not the full distribution. (2) *Top causal factors are robust across models*. We run ablation with GPT-4o-mini and GPT-5.4 for posterior attribution: for items where GPT-5.4 assigns > 0.9 attribution, GPT-4o-mini agrees 85.8% of the time. The two models diverge on the tail but agree on dominant factors. (3) *Tail noise does not corrupt training*. GPT-4o-mini assigns more mid-range scores (31.4% vs. 12.7%), but training signal is dominated by top-end agreement. Training with cleaner GPT-5.4 attributions could further improve performance, left for future work.
>
> | Metric | GPT-4o-mini | GPT-5.4 |
> |---|---|---|
> | Mean Score | 0.158 | 0.102 |
> | % Zero (< 0.05) | 54.4% | 72.2% |
> | % Mid (0.1–0.5) | 31.4% | 12.7% |
> | % Top (≥ 0.9) | 3.3% | 1.4% |
> | Entropy | 3.004 | 2.735 |
>
>
> **[Additional time-series baselines]** We add ChatTime-Base [1], ChatTime-Chat [1], and From News to Forecast [2] as additional event-driven time series baselines. These methods jointly leverage time series data and event/news data. We also add Informer and Reformer as additional time-series baselines.
>
>
> | Method | Polymarket MAE ↓ | Polymarket RMSE ↓ |
> |---|---|---|
> | Autoformer | 0.1476 | 0.1182 |
> | DLinear | 0.1100 | 0.0815 |
> |Informer|0.0703|0.0979|
> |Reformer|0.0687|0.0959|
> | TimeLLM | 0.0800 | 0.0612 |
> |ChatTime-Base | 0.1000 | 0.1413|
> |ChatTime-Chat|0.0883|0.1275|
>  |From news to forcast| 0.0634 | 0.0782|
> | SWM | 0.0649 | 0.0560 |
>
> **[Additional frontier model baselines]** Besides GPT-5.2, we also include frontier models like Deepseek-V3 and R1. But these models have lower performance compared with GPT-5.2.
>
> | Model | MAE ↓ | RMSE ↓ |
> |---|---|---|
> | SWM | 0.0649 | 0.0560 |
> | GPT-5.2 | 0.0587 | 0.0522 |
> | DeepSeek-V3 | 0.1069 | 0.1615 |
> | DeepSeek-R1 | 0.1752 | 0.2332 |
>
> **[Null event ablation]** We ablate the null event design by comparing two settings: (1) with null event (default)—when no news is confidently attributed, the model assigns probability to a dedicated null event, explicitly representing "no meaningful news drove this change," and (2) without null event—when no news is confidently attributed, the model can only fall back to a uniform distribution over all candidates. We train a 0.6B social world model and social attributor for each setting.
>
> | Metric | With Null Events (baseline) | Without Null Events (ablation) | Diff |
> |---|---|---|---|
> | MSE | 0.01774 | 0.01829 | +3.1% ↑ |
> | MAE | 0.0751 | 0.0803 | +6.8% ↑ |
>
>
> **[Event description ablation]** By default, each news event is represented by its title and a 300-character truncated description. We ablate this by comparing: (1) title only, vs. (2) title + description (default). We evaluate on the breakpoint big-change subset of Kalshi, where news-driven effects are most pronounced, to make differences visible.
>
> Posterior Attribution Difference (w/ vs. w/o Description)
>
> | Metric | Value |
> |---|---|
> | Spearman Corr | 0.351 |
> | Kendall Tau | 0.365 |
> | Top-1 Agreement | 73.9% |
> | Top-3 Overlap | 84.1% |
> | Score Diff (Desc − Title) | +0.045 |
>
> Forecasting Results with Different Event Description
>
> | Threshold | Title MAE | Title+Desc MAE | Title RMSE | Title+Desc RMSE |
> |---|---|---|---|---|
> | All | 0.1591 | 0.1569 | 0.2196 | 0.2174 |
> | ≥ 0.1 | 0.1591 | 0.1569 | 0.2196 | 0.2174 |
> | ≥ 0.5 | 0.1522 | 0.1494 | 0.2096 | 0.2060 |
> | ≥ 0.7 | 0.1467 | 0.1426 | 0.1998 | 0.1948 |
> | ≥ 0.8 | 0.1375 | 0.1339 | 0.1840 | 0.1799 |
> | ≥ 0.9 | 0.1553 | 0.1507 | 0.2088 | 0.2032 |
> | ≥ 0.95 | 0.1723 | 0.1661 | 0.2309 | 0.2233 |
>
>
> **[Typos]** Thanks for pointing out the relevant typo in Line 263. We will correct them in camera-ready version once got accepted.
>
> [1] Wang et al. ChatTime: A Unified Multimodal Time Series Foundation Model Bridging Numerical and Textual Data.
>
> [2] Wang et al. From News to Forecast: Integrating Event Analysis in LLM-Based Time Series Forecasting with Reflection.

---

> > ### Author Rebuttal · Reviewer_WDi2 · 2026-04-01
> >
> > Thank you for addressing the comments! I will raise my score.
> >
> > I think it would be useful context to include supporting evidence of price changes being influenced by 1-5 causal factors, if possible (via small ablation studies with more lenient attributions or relevant citations). The comparison provided above indicates that GPT-5.4 sharpens the distribution of attributions, and it would be helpful context/supporting evidence that would provide a rough indication of the benefits of scaling the posterior attribution model.

---

> > > ### Author Response · Authors · 2026-04-08
> > >
> > > Thank you for the helpful suggestion, and we also sincerely appreciate your recognition of our work. We agree that it would strengthen the paper to provide more supporting evidence on whether belief shifts are typically driven by a small number of salient factors. We will add this context in the revision, either through a small ablation with more lenient attribution settings and/or through relevant citations, to give a rough indication of the benefits of scaling the posterior attribution model.

---

### Decision · Program_Chairs · 2026-04-30

**Decision:**

Accept (regular)

**Comment:**

The paper introduces a modeling framework called “Social World Model” to model the dynamics of social beliefs and how they change in response to world events. The model is probabilistic in nature and learned with a variational approach. A new benchmark, SWM-Bench, is introduced for evaluation. It consists of prediction market data. SWM is shown to outperform traditional time series and news modeling frameworks for prediction tasks on SWM-Bench.

Reviewers appreciated the contributions of the paper. Overall, they felt the problem was clearly outlined, the methods were well motivated and appropriate, and good evidence was given for their effectiveness. The contribution of a benchmark was considered important, since this is an active/emerging area of research with limited benchmark data for evaluation and comparison. Weaknesses and questions focused on specific aspects of the framework such as details of posterior attribution and additional baselines and ablations. Overall, these were well addressed by the rebuttal. One reviewer (6cjR) had a more negative overall assessment, but their primary concern was about the scope of the paper: that it focused primarily on market and news data as opposed to the much broader set of social issues. Other reviewers find the current scope acceptable and a worthwhile contribution to an emerging area.